# Sustained Compression with a Pneumatic Cuff on Skeletal Muscles Promotes Muscle Blood Flow and Relieves Muscle Stiffness

**DOI:** 10.3390/ijerph19031692

**Published:** 2022-02-01

**Authors:** Masaaki Nakajima, Tomoka Tsuro, Akemi Endo

**Affiliations:** 1Department of Physical Therapy, School of Health Science and Social Welfare, Kibi International University, Takahashi City 716-9508, Japan; sjstarry.sky1011@gmail.com; 2Medical Concierge Co., Ltd., Okayama City 700-0024, Japan; akemi050995@gmail.com

**Keywords:** muscle blood flow, sustained compression, pneumatic cuff, near-infrared spectroscopy

## Abstract

(1) Purpose: This study aimed to examine whether a pneumatic cuff could promote muscle blood flow and improve muscle stiffness by continuously compressing muscles with air pressure in healthy college students. (2) Method: Twenty-one healthy collegiate students participated in this study. The probe of the near-infrared spectrometer was attached to the upper surface of the left gastrocnemius muscle, and a cuff was wrapped around the left lower leg. The cuff was inflated to 200 mmHg. After 10 min, the cuff was deflated, and the patient rested for 10 min. Muscle stiffness and fatigue were assessed before and after the intervention. (3) Results: During 10 min of continuous compression, StO_2_ continued to decrease until seven min of compression. After 10 min of continuous compression, StO_2_ was 30.8 ± 10.4%, which was approximately half of 69.2 ± 6.1% at rest. After the release of the pneumatic cuff compression, the StO_2_ remained higher than that at rest from 1 to 10 min. Muscle hardness was 19.0 ± 8.0 before intervention was 8.7 ± 4.8 after the intervention. Muscle fatigue was 6.6 ± 1.7 cm before the intervention and 4.0 ± 1.6 cm after the intervention. (4) Conclusions: This study suggests that sustained muscle compression using a pneumatic cuff can promote muscle blood flow and improve muscle stiffness and fatigue.

## 1. Introduction

CO_2_ is known as a vasodilator and has been clinically applied as an artificial carbon dioxide spring bath. When the human body is immersed in a carbon dioxide spring, the CO_2_ that enters the skin transcutaneously relaxes the vascular smooth muscle of the skin and promotes blood flow [1,2] For this reason, artificial carbon dioxide baths have excellent therapeutic effects on bedsores and skin ulcers [3,4]. As the vasodilatation effect of artificial carbon dioxide springs is a concentration gradient effect [5], CO_2_ that enters percutaneously has a great effect on relaxing the vascular smooth muscles in the shallow skin layer and promoting blood flow. However, its effect on the muscles in the deeper layers is weakened because the concentration of CO_2_ is reduced by blood flow before it reaches them. A method to effectively apply this vascular smooth muscle relaxing effect of CO_2_ on blood vessels in muscle was sought. Then, a method was developed to utilize the CO_2_ generated by the metabolism of muscle cells to increase the CO_2_ concentration in the muscle and relax the vascular smooth muscle by keeping the muscle under restricted blood flow for a certain period (Figure 1). This method involves manually applying pressure to the muscle and holding it there for a period of time. Nakajima reported that when this new method was applied to patients with OA of the knee, it promoted muscle blood flow and provided excellent pain relief [6].

On the other hand, Knee OA affects a large number of people [7]. Furthermore, the number of affected people is increasing due to the growing aging population in many developed countries [8,9]. It can be challenging to treat a large number of people with knee OA using manual treatment strategies. To solve this problem, it would be beneficial to develop a device-based treatment method that does not rely on human resourcing.

This study aimed to develop a pneumatic continuous compression therapy for knee OA pain. As a pilot study, we first examined the possibility of promoting muscle blood flow by continuously compressing the muscle with air pressure using a self-made pneumatic cuff in healthy university students.

## 2. Materials and Methods

Settings of the air pressure value of the pneumatic cuff and the pressure holding time: In this study, the pneumatic cuff is required to have sufficient muscle blood flow restriction. The pressure of the pneumatic cuff was set at 200 mmHg [10], which is sufficient to restrict blood flow in the lower extremities. In the blood flow restriction method for the extremities, it is said that the decrease in oxygen concentration is no longer observed about 6 min after the start of blood flow restriction [11,12], and at this time, the muscles are considered to be in a state of functional anoxia. It is believed that carbon dioxide concentration reaches a high level due to myocyte metabolism when functional anoxia occurs under restricted muscle blood flow. Therefore, the blood flow restriction of the leg muscles in this study was also set to a blood flow holding time of 10 min, including 6 min during which functional anoxia was reached.

Evaluation of muscle blood flow: Near-infrared spectrometer was used to evaluate muscle blood flow. It is used as a non-invasive method to measure changes in blood circulation in muscles. Muscle tissue oxygen saturation (StO_2_), which can be measured by near-infrared spectroscopy, reflects the balance between oxygen supply and consumption in muscle tissue [13] and can capture changes in muscle blood flow at the local muscle tissue level [14,15,16].

Experimental protocols: Twenty-one healthy collegiate students from Kibi International University participated in this study. The subjects’ average age was 20.3 ± 0.9 years. Their average body mass index was 22.0 ± 4.0. Each subject provided informed consent to participate in the study according to the procedure of the Institutional Review Board of the Kibi International University Research Ethics Committee. After consent was provided, each subject was asked to complete a medical history questionnaire. Subjects were excluded from the study if they had any history of back or lower limb problems. The subjects were asked to present in the evening after their lectures when their legs were tired to some extent. After 10 min of rest in a chair, the subject was placed in a supine position, the probe of the near-infrared spectrometer was attached to the upper surface of the gastrocnemius muscle on the posterior surface of the left lower leg, and a cuff was wrapped around the lower leg. The cuff has an air chamber that can be inflated by a pump to compress the gastrocnemius muscle (Figure 2). After 1 min of rest, the cuff was inflated to 200 mmHg, and the gastrocnemius muscle was compressed. After 10 min, the cuff was deflated to release the gastrocnemius muscle compression, and the patient was allowed to rest in that position for 10 min. During this period, oxy-Hb and deoxy-Hb of the left lower gastrocnemius muscle were measured using a muscle infrared spectrometer (Omega Wave BOM-L1 TR), and the data were stored on a personal computer using an A/D converter Power Lab. From the data obtained, StO_2_ was calculated every minute from the start. Muscle stiffness and fatigue were assessed before and after the intervention. The ASKER DUROMETER (Kobunshi Keiki Co., Ltd., Kyoto, Japan) was used to evaluate muscle stiffness. The visual analog scale was used to assess muscle fatigue.

Data analysis: A one-way analysis of variance with repetition was performed on StO_2_ from rest to 10 min after the start of compression. The Bonferroni method was used for multiple analyses. A one-way analysis of variance with repetition was performed for StO_2_ at rest and every minute after pressure release for 10 min, and the Dunnett method was used for multiple comparisons at rest and after pressure release. All data are presented as the mean ± SD. The level of statistical significance was set to α = 0.05. Stat View (software version 5.0) was used for the statistical analysis.

## 3. Results

Continuous compression of the gastrocnemius muscle at 200 mmHg for 10 min was performed using a pneumatic cuff. During this period, StO_2_ continued to decrease until 7 min of compression and plateaued after 8 min of compression. StO_2_ just before the release of compression (at the time of 10 min of continuous compression) was 30.8 ± 10.4%, about half of the resting rate (69.2 ± 6.1%). After the pressure was released by the air cuff, the StO_2_ remained higher than at rest from 1 to 10 min (Figure 3). Muscle stiffness was 19.0 ± 8.0 and 8.7 ± 4.8 before and after the intervention, respectively. Muscle fatigue was 6.6 ± 1.7 cm before the intervention and 4.0 ± 1.6 cm after the intervention.

During 10 min of continuous compression, StO_2_ decreased until the 7-minute time point. After pressure release, StO_2_ was higher than at rest at all time points from 1 to 10 min.

## 4. Discussion

StO_2_ continued to decrease until the 7-minute point of compression, after which it plateaued. The results were similar to those reported by Hamaoka et al. [11] and Sahlin et al. [12]. The StO_2_ after pressure release was 77.2 ± 3.1% after 1 min of release, which was about 10% higher than the resting value of 69.2 ± 6.1. Even after 10 min of opening, StO_2_ remained higher than at rest at 74.3 ± 5.5%. In Nakajima’s manual compression therapy, StO_2_ at rest was 60.1 ± 5.7%, and after the intervention, StO_2_ was 65.3 ± 4.8%—an increase of about 10% [6]. Intervention with the pneumatic cuff is thought to have the same level of effect on promoting muscle blood flow as manual compression therapy by Nakajima. Its blood flow promoting effect was shown to be effective for 10 min.

Static stretching (SST) is widely practiced to promote recovery from muscle fatigue and to improve exercise performance by increasing muscle and tendon flexibility. SST is a method of slowly stretching the target muscle group without a reactionary movement and holding the muscle stretch. The physiological effect of SST is known to be the enhancement of blood circulation. There are reports that SST was performed, and muscle oxygen saturation was evaluated. Nagasawa et al. [17] applied an SST to the forearm flexor muscle group by means of passive dorsiflexion of the wrist joint and evaluated StO_2_. StO_2_ was 69.2 ± 2.2% at rest and 63.5 ± 2.0% at 30 s after the start of SS, and promptly increased to a maximum value of 72.5 ± 1.8% at 19.8 ± 5.5 s after the end of SST. The highest StO_2_ value in this report would be about 5% higher than at rest. The same rate of increase was reported in a report that assessed StO_2_ after SST of the triceps femoris muscle [18]. In our intervention, using the pneumatic cuff, StO_2_ increased by about 10%, which suggests that the effect of promoting muscle blood flow is greater than that of SST. The increase in muscle blood flow in SST may be due to a temporary restriction of blood flow in the muscle during SST. A temporary restriction of blood flow causing an increase in blood flow after release is known as reactive hyperemia. The excellent blood flow-promoting effect of the pneumatic cuff is due to this reactive hyperemia as well as the effect of CO_2_. Blood flow to the muscular tissue capillary network is controlled by the anterior capillary sphincter (vascular smooth muscle), which has some dormant capillary network. Intervention with a pneumatic cuff is expected to relax the anterior capillary sphincter muscle, allowing blood to circulate through all capillary networks in the muscle and flush out pain-related and fatigue substances.

Increased blood flow also increases shear stress stimulation of vascular endothelial cells, which releases NO and relaxes vascular smooth muscle [19]. The intervention with the pneumatic cuff was considered to be synergistic in the above effects, resulting in a high StO_2_ value even after 10 min, i.e., a state of accelerated blood flow.

Other therapies using pneumatic cuffs include intermittent pneumatic compression therapy for the prevention of edema and deep vein thrombosis [20]. The inflation pressure is 30 to 60 mmHg in the upper extremities and 40 to 80 mmHg in the lower extremities, with an inflation time of 80 to 100 s, a contraction time of 25 to 35 s, and a treatment time of 2 to 3 h. No effect on muscle blood flow has been reported.

This study suggested that applying sustained muscle compression with 200 mmHg air pressure using a pneumatic cuff for 10 min promoted muscle blood flow for at least 10 min and improved muscle stiffness and fatigue. In the future, we will verify the effect of promoting muscle blood flow at different pressures and holding times. Then, we would like to apply it to actual cases of knee OA to verify its effectiveness in reducing pain.

## 5. Conclusions

The purpose of this study was to evaluate whether continuous compression of muscles using a pneumatic cuff for a certain period of time promotes blood flow after release, using a near-infrared spectrometer. While the pneumatic cuff pressure was maintained at 200 mmHg, StO_2_ in the muscle continued to decrease until 7 min after the start of compression, indicating suppression of blood flow. After the release after 10 min, StO_2_ increased more than before the intervention, indicating an increase in blood flow. The promotion of blood flow continued even after 10 min of release. Muscle stiffness was reduced after the intervention. It was found that sustained compression with a pneumatic cuff could promote blood flow in the muscle and relieves muscle stiffness. Next, we would like to clinically apply the continuous compression method using a pneumatic cuff for myogenic pain in knee osteoarthritis.

## Figures and Tables

**Figure 1 ijerph-19-01692-f001:**
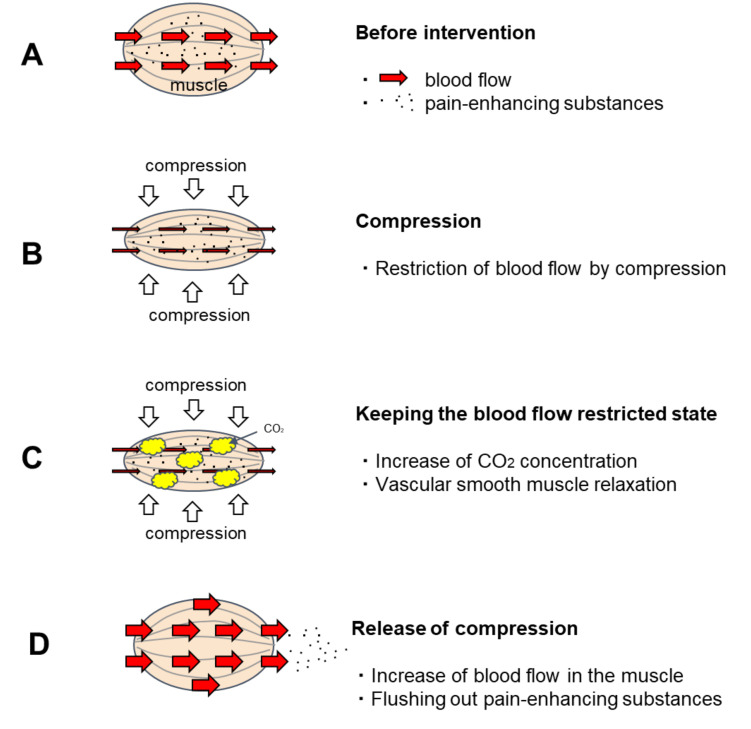
Predicted effect of temporary blood flow restriction on blood flow. (**A**). Before intervention: Pain-enhancing substances are stored in the muscles. (**B**). Compression: The blood vessels in the muscle are squeezed by the pressure, restricting blood flow. (**C**). Keeping the blood flow restricted state: When keeping the blood flow restricted state, the intramuscular CO_2_ concentration increases due to CO_2_ generated by the metabolism of muscle cells. CO_2_ has the effect of relaxing vascular smooth muscle. When CO_2_ concentration rises sufficiently, the vascular smooth muscle relaxes. (**D**). Release of compression: Blood flow is accelerated after the pressure is released because the vascular smooth muscle is relaxed. As the vascular smooth muscle relaxes, blood also circulates through to the resting capillary network. Hence, pain-enhancing substances in the muscles are flushed out.

**Figure 2 ijerph-19-01692-f002:**
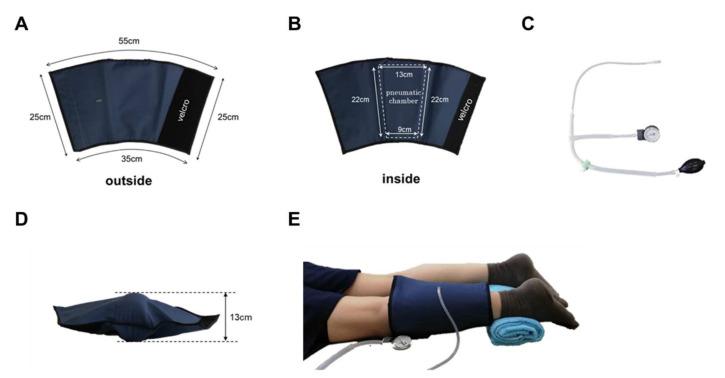
Pneumatic cuff. The cuff was made of sturdy nylon and was sized to cover the lower leg. Velcro makes it easy to put on and take off (**A**,**B**). It has an internal air chamber that compresses the gastrocnemius muscle (**B**). Air is inserted through the valve on the surface of the cuff (**A**). The pump was designed with a pressure gauge that can be used to inflate the air to a specified pressure (**C**). When the pneumatic cuff is filled with air (200 mmHg) when not worn, the thickest part is 13 cm (**D**). The subject was placed in a prone position, and a pneumatic cuff was wrapped around the left lower leg (**E**).

**Figure 3 ijerph-19-01692-f003:**
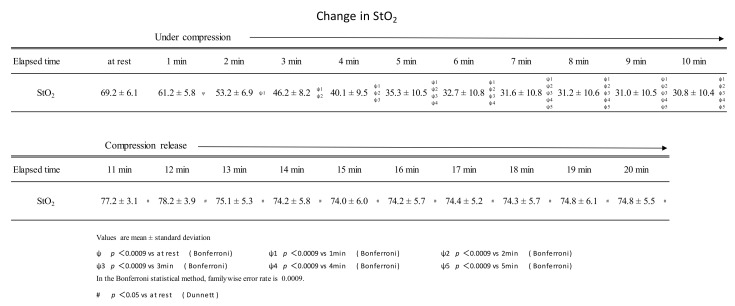
Changes in StO_2_ during sustained compressions and after the release of compressions.

## Data Availability

All data are available upon request.

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
