# Peer review of "Sustained Compression with a Pneumatic Cuff on Skeletal Muscles Promotes Muscle Blood Flow and Relieves Muscle Stiffness"

_ijerph, 2022, doi:10.3390/ijerph19031692_

Round 1

Reviewer 1 Report

Thank you for the opportunity to review a interesting paper addressing current patient needs.

  1. Introduction

In the introduction, the authors describe osteoarthritis of the knee. The study was conducted only on young, healthy individuals. Additionally, in the introduction, the authors have used references to quite “old” publications.

The introduction has an abnormal structure. First the authors write about osteoarthritis of the knee, then about the effect of CO2, and then again about osteoarthritis. The question is - what for?

“This study aimed to develop a pneumatic continuous compression therapy for knee OA pain. First, we examined the possibility of promoting muscle blood flow by continuously  compressing the muscle with air pressure using a self-made pneumatic cuff in healthy university students”

The aim is incorrect.

  1. Results

It might be useful to provide detailed results in a table. The description of statistically significant results is missing from the text. They are only found in the graph!

  1. Discussion

The authors did not indicate why only 200 mmHg pressure was used in the study. Additionally, in the discussion, the authors have used references to quite “old” publications.

It would be useful to discuss the results with more publications by other authors.

  1. Conslusions

Results are not the place to repeat results.  These items should not be repeated.

Author Response

Introduction

In the introduction, the authors describe osteoarthritis of the knee. The study was conducted only on young, healthy individuals. Additionally, in the introduction, the authors have used references to quite “old” publications.

[This study is a pilot study to investigate whether pneumatic cuffs can promote muscle blood flow. In the next study, we would like to apply it to actual patients with knee osteoarthritis.

Recent papers have been added to the references. ]

The introduction has an abnormal structure. First the authors write about osteoarthritis of the knee, then about the effect of CO2, and then again about osteoarthritis. The question is - what for?

[The purpose of pneumatic cuff therapy is to relieve the pain of knee osteoarthritis.

Pneumatic cuff therapy is based on the principle of action of manual compression therapy.

The principle of action of manual compression therapy is the flushing out of pain-related substances and fatigue substances in the muscle due to the relaxing effect of CO2 on vascular smooth muscle.

Therefore, I started writing from the concept of action of manual compression therapy.]

“This study aimed to develop a pneumatic continuous compression therapy for knee OA pain. First, we examined the possibility of promoting muscle blood flow by continuously compressing the muscle with air pressure using a self-made pneumatic cuff in healthy university students”

The aim is incorrect.

[Before examining whether or not the pneumatic cuff can promote muscle blood flow in middle-aged and elderly people with actual knee osteoarthritis, we examined it in young healthy people.

It is meaningful to know the response of young healthy subjects to pneumatic cuff therapy.

If the response in middle-aged and elderly people with knee osteoarthritis is different from the response in young healthy people, it will be useful information for seeking effective treatment.]

Results

It might be useful to provide detailed results in a table. The description of statistically significant results is missing from the text. They are only found in the graph!

[Changed from a graph to a table.]

Discussion

The authors did not indicate why only 200 mmHg pressure was used in the study. Additionally, in the discussion, the authors have used references to quite “old” publications.

[Adequate blood flow restriction is required for therapy with pneumatic cuffs.

Therefore, based on previous studies, we decided on a pressure of 200 mmHg to sufficiently restrict blood flow.

・Kubota, A.; Sakuraba, K.; Sawaki, K.; Sumide, T.; Tamura, Y. Prevention of disuse muscular weakness by restriction of blood flow. Med Sci Sports Exerc 2008, 40(3). 529-534.]

Conslusions

Results are not the place to repeat results.  These items should not be repeated.

[I have corrected the problem as you instructed.]

Reviewer 2 Report

The authors have studied the effect of pneumatic cuff on gastrocnemius muscle in inducing pain relief. The study design is incomplete as there is no proper control in the experiment. I have following comments which needs to be addressed before its final acceptance.

  1. The authors should use either the contralateral leg or the same leg without compression as control which is missing in the experiment.
  2. The author should provide the justification or  cite any previous reference to use 200mm Hg presure in their study.
  3. The authors have mentioned about the increase in CO2 concentration  after the compression but they have not shown any data on it. 
  4. The total duration of the study is only for 10 min and the authors should provide justification on it whether this short time period will be enough to cure the patients.
  5. The authors should also consider measuring the local level metabolites such as lactate and muscle swelling parameters before and after the experiment.

Author Response

  1. The authors should use either the contralateral leg or the same leg without compression as control which is missing in the experiment.

[The resting state was used as a control, referring to previous studies using related NIRS.]

・Soares, R.N.; Inglis, E.C.; Khoshreza, R.; Murias, J.M.; Aboodard, S.J. Rolling massage acutely improves skeletal muscle oxygenation and parameters associated with microvascular reactivity: The first evidence-based study. Microvasc Res. 2020; 132: 104063.

・Owen Jeffries, Mark Waldron, John R Pattison, Stephen D Patterson. Enhanced Local Skeletal Muscle Oxidative Capacity and Microvascular Blood Flow Following 7-Day Ischemic Preconditioning in Healthy Humans. Front Physiol 2018 May 9;9:463.

・Differences in oxidative metabolism modulation induced by ischemia/reperfusion between trained and untrained individuals assessed by NIRS. Soares RN, McLay KM, George MA, Murias JM. Physiol Rep. 2017 Oct;5(19):e13384.

・Sahlin, K. Non-invasive measurements of O2 availability in human skeletal muscle with near-infrared spectroscopy. Int J Sports Med 1992, 13 Suppl 1:S157-160.

  1. The author should provide the justification or cite any previous reference to use 200mm Hg presure in their study.

[Adequate blood flow restriction is required for therapy with pneumatic cuffs.

Therefore, based on previous studies, we decided on a pressure of 200 mmHg to sufficiently restrict blood flow.]

・Kubota, A.; Sakuraba, K.; Sawaki, K.; Sumide, T.; Tamura, Y. Prevention of disuse muscular weakness by restriction of blood flow. Med Sci Sports Exerc 2008, 40(3). 529-534.

  1. The authors have mentioned about the increase in CO2 concentration after the compression but they have not shown any data on it.

[Thank you for your advice.

We have modified to mention the change in StO2.]

  1. The total duration of the study is only for 10 min and the authors should provide justification on it whether this short time period will be enough to cure the patients.

[Hamaoka et al. state that functional anoxia occurs 6 minutes after the start of muscle blood flow restriction by 200 mmHg air pressure. It is thought that carbon dioxide concentration reaches a high level due to myocyte metabolism when functional anoxia occurs under restricted muscle blood flow. Therefore, the blood flow restriction of the leg muscles in this study was also defined as a blood flow holding time of 10 min, including 6 min during which functional anoxia was reached.]

  1. The authors should also consider measuring the local level metabolites such as lactate and muscle swelling parameters before and after the experiment.

[Thank you for your advice.

I will consider it for another opportunity.]

Round 2

Reviewer 1 Report

2. Materials and Methods
Settings of the air pressure value of the pneumatic cuff and the pressure holding time
- I'm not sure if some of the information in this paragraph isn't worth moving into the discussion. 

Table 1.
- It is worth improving the readability of the table

References
The new numbering does not match because some references are split between paragraphs.

Author Response

About [Settings of the air pressure value of the pneumatic cuff and the pressure holding time]

- The reason for setting the compression holding time to 10 minutes is stated to indicate the reason. The text has been rewritten.

About [Table 1]

- The table has been revised to make it easier to read.

I would like to appreciate you for reviewing my paper during the difficult time of the corona disaster.

Reviewer 2 Report

The authors have addressed all my concerns by providing references and additional results. I support the publication of the manuscript.

Author Response

I would like to appreciate you for reviewing my paper during the difficult time of the corona disaster.